# Exhibit: Converting my PhD thesis into HTML

Damien Desfontaines
damien@desfontain.es

March 7, 2021

## 1    What is this?

I converted my PhD thesis, titled *Lowering the cost of anonymization*, into HTML; the result is available at the URL desfontain.es/thesis. The motivation and process are described in a blog post (desfontain.es/privacy/latex-to-html.html); and I was told that this work could constitute an interesting exhibit submission to the 2021 *Rethinking ML Papers* workshop at ICLR. This supplementary document completes both pages to provide more context around the submission.

## 2    What motivates this?

I did this project mostly out of frustration that all the work you put into a PhD thesis ends up in a long PDF document that nobody ever reads. This is especially the case in computer science, where PhD dissertations are often a list of papers stapled together, and individual papers typically make for a more convenient read.

When writing my thesis, I tried to avoid this common pitfall, and reused materials from my blog to make my thesis more self-contained and accessible to non-specialists. Nonetheless, the end result (available in PDF form at desfontain.es/thesis.pdf) is still, by its very form, intimidating and borderline hostile to readers. I discuss this further in the submitted blog post, where I make the case for HTML as a better option.

During this process, I bumped into a number of unexpected difficulties.

## 3    Why is this interesting?

I am hoping that presenting this exhibit at the *Rethinking ML Papers* would encourage discussions along at least three axes.

1. The benefits of HTML over (or in addition to) PDF as a format for scientific texts, and more generally to disseminate scientific information.

2. The technical difficulty of generating HTML pages from LaTeX source code, the different possibles approaches to doing so, and how we can improve the situation.

3. The social conventions around PhD dissertations in computer science, and how .

# 4   What are the accessibility implications?

Though non-trivial, it is much easier to build accessible HTML pages rather than PDF files, especially for documents that mostly contain text. Simple responsive design can allow users to zoom to arbitrary levels without sacrificing the ability to browse comfortably. Semantic elements can separate header, footer, and navigation panels from content, making it easier for screen readers to convey the information to their users.

Though I am far from an expert, I have tried to follow standard accessibility best practices on my website: using semantic elements, keeping things simple, manually indicating to screen readers where the content starts, avoiding low-contrast colors, making sure that the pages display well at any zoom level.

There is certainly room for improvement. For examples, figures in a scientific paper have a caption and their content is discussed in the text, but does that compensate for the lack of alternate text in the HTML generated from LaTeX? MathJax, used to render equations, claims to be compatible with screen readers, but how well does it work in practice? If you have concrete suggestions on how to improve accessibility on my website, I'd love to hear about it and make things better!

