# OpenReview forum: "Exhibit: Converting my PhD thesis to HTML"
_ICLR.cc/2021/Workshop/Rethinking_ML_Papers — Rethinking ML Papers - ICLR 2021 workshop Poster_

### Official Review · AnonReviewer1 · 2021-03-25
**Clearly written, but I wish more detail was provided**

**Accessibility:**

Score of 3 (Neutral): Submission proposes methods to improve accessibility, but the level of intended accessibility is not well-articulated. Also, the limitations and exceptions are not stated.

**Groundsforrejection:**

This submission breaks double blind as the author is named in the pdf submission.

**Litreview:**

Score of 2 (Needs Improvement): The submission leaves out prominent examples of previous work in the area.

**Problemstatement:**

Score of 3 (Neutral): The submission states the problem, but either addresses a different one or fails to address the problem in a reasonably-significant manner (Note: as this is a new workshop, we’d like to be lenient with the word significant)

**Relevance:**

Score of 3 (Neutral): Attempt was clearly made to address a theme of the workshop, but it seems that the work was ‘retrofitted’ to match the theme of the workshop.

**Results:**

Score of 3 (Neutral): Submission is well designed and provides a good level of coherency/novelty/interactivity.

**Reviewerconfidence:**

Based on my personal research, professional experience, and technical contributions in the area of open science, I am highly confident in this assessment.

**Reviewtext:**

The author has made a webpage of their thesis.  They clearly explain the problem and how it was done.  At the same time, I wish the problem was further motivated.  No citations are provided, for example, which limits the ability of this work to be connected to other works that try to digitize academic papers.  Moreover, the author does not address the specific question why simply converting the original text to HTML is sufficient to enabling researchers to engage with dissertations.  The author notes that dissertations are less often read than academic papers, but the author does not demonstrate why HTML alone is sufficient to make this change.  For example, other researchers suggest that alternate forms of media may be better to elicit curiosity in a dissertation topic, see:

https://journals.sagepub.com/doi/10.1177/1357034X11430965

Moreover, digital journals have attempted to take advantage of publication by embedding media such as code into digital publications, such as:

https://elifesciences.org/labs/d42fe2b9/integrating-binder-and-stencila-the-building-blocks-to-increased-open-communication-and-transparency

I believe that this contribution is a good step towards improving accessibility of dissertation research, but would appreciate also more technical details regarding the process.  The authors do not describe how the website was built, nor if the authors made a novel library to create this website, so that others could apply these methods on their own papers as in:

https://jupyterbook.org/intro.html

Overall, I am not against acceptance of this work, but I hope these details will be provided at the workshop.  I am concerned however regarding the fact the author includes their name in the submission, which would break double blind, which I believe would be reasonable cause for desk rejection should the organizers wish.



**Score:**

Accept: The reviewer believes the submission provides a novel and reliable scheme to improve science communication but needs improvement.

---

### Official Review · AnonReviewer2 · 2021-03-30
**Review on exhibit converting LaTeX source code to HTML**

**Accessibility:**

Score of 3 (Neutral): Submission proposes methods to improve accessibility, but the level of intended accessibility is not well-articulated. Also, the limitations and exceptions are not stated.

**Litreview:**

Score of 3 (Neutral): The submission acknowledges previous work, but does not necessarily explain how the submission differentiates itself (i.e we want to avoid the “deluge of citation” strategy, leaving the reviewer to click through references and figure this part out for themselves).

**Problemstatement:**

Score of 4 (Strong): The submission sets a very strong example of how to address the problem, which should be relevant to the workshop themes.

**Relevance:**

Score of 2 (Needs Improvement): Submission attempts to address a theme of the workshop, but either misunderstands the theme or only superficially addresses it (i.e brings it up in an introduction, but does not attempt to address it or connect it to the submission)

**Results:**

Score of 2 (Needs Improvement): Submission shows a poor level of clarity, novelty, coherency, and interactivity.

**Reviewerconfidence:**

4 -- I am fairly confident with my review of the submission and my analysis of where it is lacking.

**Reviewtext:**

Summary:
The author presents their work to convert a document written in LaTeX to HTML pages. The submission presents conversion of the author's Ph.D. thesis to a series of web pages. The exhibit material, specifically the blog post, is well-written and clear. The proposed method is not particularly novel but has the potential to drive conversations around making academic content more accessible to a broad audience.

Pros:
* The author walks readers through errors that arose during their LaTeX to HTML conversion and includes relevant links on how they resolved it.
* This approach appears to be applicable as a starting point to convert any LaTeX source code to HTML.
* The use of an identifier to distinguish sections that could appeal to an audience who may not want to go into the specifics/math is neat!

Cons:
* The submission could have touched upon future directions/avenues of development to generalize. Having the author's perspective on improvements beyond resolving errors would have been interesting.
* In section 3 of the document (“Why is this interesting?”), the author seems to have missed completing their thought. Reviewing & updating this would communicate ideas clearly.

Feedback/Suggestions:
* Sub-headings are about the same size as the rest of the text and hard to read. Eg- Section 2.1.1: “Policy questions raised by k-anonymity” is not obvious.
* Given the author tried 3 other tools in addition to TeX4ht, a short note on how pandoc, lwarp and LaTeXML failed, would allow readers to explore further instead of directly turning these down as potential tools of choice.
* For hyperlinks to definitions, references, or other material, it might make sense to have the content/partial content/title appear when we hover over the hyperlink.
* Include direct links to access papers in the bibliography. A long list of papers on a webpage would be less interesting than even a similar list in a PDF.
* Text to move between pages/Table of Contents is hard to read.

**Score:**

Accept: The reviewer believes the submission provides a novel and reliable scheme to improve science communication but needs improvement.

---

### Author Response · Authors · 2021-05-07
**Some links!**

- Thesis in HTML form: https://desfontain.es/thesis
- Blog post describing the process: https://desfontain.es/privacy/latex-to-html.html
- Poster: https://desfontain.es/PDFs/PosterRethinkingMLPapers2021.pdf

---

### Meta-Review · Area_Chair1 · 2021-04-01

**Recommendation:** Accept
**Confidence:** 4

**Metareview:**

This exhibit describes the rationale behind converting a PhD thesis to an HTML page, and presents a first-person account of the process. Both reviewers appreciated this idea and found the submission to be "clear". However, reviewers also note that there weren't any references to prior efforts that undertook a similar initiative. Furthermore, one reviewer sees limited value in this as solely an 'exhibit' if the associated code weren't released (or minimally, enough details to reproduce the exhibit weren't provided). Having carefully read through the submission, I agree with the reviewer assessment.
I believe that presenting this exhibit at the workshop would a) expose the ML audience to the idea of hosting a thesis as a webpage, and b) allow the author(s) of the exhibit to elicit meaningful feedback in improving their exhibit (particularly from a pedagogy and accessibility standpoint). I strongly encourage the authors to include more details about the process (and code) when preparing the camera ready version. I also urge the authors to voice the scope to which this format is accessible. Further, the eventual submission must also discuss existing packages or efforts to convert papers / LaTeX to web formats. (e.g. https://learning-from-play.github.io/ is a CoRL 2020 paper that was converted to a webpage, https://github.com/latex2html/latex2html is a LaTeX to HTML translator; there may be more efforts that I might have missed out on).
Note about double-blind violation: The program chairs deduce that the authors intended to submit an exhibit, but have inadvertently submitted to an incorrect track. In the eventual workshop proceedings, we will list this submission as an "exhibit / workflow".

---

### Decision · Program_Chairs · 2021-04-01

Accept (Poster)